# Alkaloid Profile in Wild Autumn-Flowering Daffodils and Their Acetylcholinesterase Inhibitory Activity

**DOI:** 10.3390/molecules28031239

**Published:** 2023-01-27

**Authors:** Julia Lisa-Molina, Pedro Gómez-Murillo, Irene Arellano-Martín, Carles Jiménez, María L. Rodríguez-Escobar, Luciana R. Tallini, Francesc Viladomat, Laura Torras-Claveria, Jaume Bastida

**Affiliations:** 1Departament de Biologia, Sanitat i Medi Ambient, Facultat de Farmàcia i Ciències de l’Alimentació, Universitat de Barcelona, Av. Joan XXIII 27-31, 08028 Barcelona, Spain; 2Independent Researcher, C/Caridad 8, Planta 2, Puerta 8, 29680 Estepona, Spain; 3Independent Researcher, C/Doctor Nicasio Benlloch, 24, 28-A, 46035 Valencia, Spain

**Keywords:** Amaryllidaceae alkaloids, *Narcissus*, acetylcholinesterase, Alzheimer’s disease

## Abstract

Amaryllidaceae alkaloids are secondary metabolites with interesting medicinal properties. Almost every *Narcissus* species can synthesize them and constitute an excellent source for their isolation and study. Several Amaryllidaceae alkaloids have shown acetylcholinesterase inhibitory activities and are a promising tool for treating cholinergic disorders such as Alzheimer’s disease (AD). Indeed, three of the four palliative treatments approved for AD are acetylcholinesterase (AChE) inhibitors and one of them, galanthamine, is an Amaryllidaceae alkaloid itself. This molecule is currently isolated from natural sources. However, its production is insufficient to supply the increasing demand for the active principle. Our main aim is to discover tools to improve galanthamine production and to prospect for potential new and more efficient drugs for AD treatment. Furthermore, we seek to broaden the knowledge of plants of the genus *Narcissus* from a chemotaxonomic perspective. Hence, in this study, we evaluate the alkaloid content through GC–MS and the AChE inhibitory activity of ten autumn-flowering *Narcissus*, which have been less studied than their spring-flowering counterparts. A total of thirty Amaryllidaceae alkaloids have been found, twenty-eight properly identified. Two *Narcissus* contained galanthamine, and seven were able to inhibit AChE.

## 1. Introduction

Alzheimer’s disease (AD) is a multifactorial neurodegenerative disorder mainly characterized by changes in memory, judgement, behavior, emotions, and abstract thinking, that ultimately interferes with physical control over the body [1]. The disease is currently affecting 37 million people, and the number is expected to rise to 50 million in 2030 and 150 million in 2050. Moreover, it is the leading cause of dementia (60–70% of the cases), which constitutes the seventh most common cause of death worldwide [2].

Even though AD has been studied for decades, its underlying cause remains unknown and only four drugs for its symptomatic treatment are currently approved. The cholinergic hypothesis is one of the main hypotheses that explain the pathophysiological manifestations of the disease. It suggests that cholinergic neurotransmission is likely to play a vital role in memory, learning, concentration, and other advanced neural functions that are highly affected in AD patients [3]. AChE inhibition therapies such as donepezil, rivastigmine, and galanthamine are widely used as symptomatic treatments for AD [4].

Galanthamine appertains to the Amaryllidaceae alkaloids, a broad and still expanding group of biogenetically related compounds with numberless bioactive properties [5,6]. This group is a distinctive chemotaxonomic feature of the Amaryllidoideae subfamily and thus is named after their former designation (Amaryllidaceae) [7]. Mainly due to the structural complexity of galanthamine, its chemical synthesis and heterologous expression are rather complex and expensive. Hence, the pharmaceutical industry currently isolates this active principle from *Narcissus* cv Carlton, *Leucojum aestivum*, and *Hippeastrum papilio* [8,9]. Nevertheless, natural cultivars impose problems such as unsuccessful cultivation or slow regeneration that limit the production yield and hinder reaching the increasing pharmaceutical demands [10]. Consequently, improving galanthamine production and finding new active principles for treating AD are of utmost importance. The AChE inhibitory potential of a large amount of Amaryllidaceae alkaloids remains to be explored.

The genus *Narcissus* (daffodils as a common name) contributes to approximately 80 out of the 850 Amaryllidoideae species, which are endemic to the Mediterranean region [11]. They grow in varied habitats, including grassland, woods, scrub, rocky crevices, and riverbanks. Most of them are synanthous (both leaves and flowers emerge at the same time), spring-flowering, and require a cold period before anthesis. The flowering date is generally dependent on spring temperatures being sufficiently high for growth [11]. However, several species are less dependent on temperature fluctuations. Such is the case of *Narcissus papyraceous*, *Narcissus bulbocodium*, and *Narcissus blancoi,* whose growth and anthesis can occur both before and after winter. In the present paper, they will be classified as facultative autumn-flowering *Narcissus* (FAF). Some other species, such as *Narcissus obsoletus*, *Narcissus deficiens*, *Narcissus serotinus*, *Narcissus cavanillesii*, *Narcissus elegans*, and *Narcissus viridiflorus*, strictly flower in autumn and are generally hysteranthous (leaves emerge at a different time than flowers). From now on, they will be addressed as strictly-autumn-flowering *Narcissus* (SAF).

Maybe due to the size and more unappealing appearance of the autumn flowering species, they have not been extensively studied and horticulturally exploited yet. Nevertheless, they have enormous potential, encompassing horticultural purposes (for instance, extending flowering periods) and pharmaceutical aims (such as improving galanthamine production or discovering new active principles). Regarding these, the natural populations of the autumn-flowering daffodils listed in Table 1 will be studied in the present project.

This study aims to find possible new sources for the improvement of galanthamine production (through biotechnological or classical breading means) and expand the knowledge of autumn-flowering daffodils by determining and evaluating their alkaloid content through a GC–MS analysis. Furthermore, it is intended to prospect possible new treatments for AD other than galanthamine by establishing the AChE inhibitory activity of each alkaloid extract. Lastly, it is sought to identify *Narcissus* species with alkaloids of interest for future studies and/or industrial exploitation.

## 2. Results and Discussion

### 2.1. Alkaloid Profiling

The bulb extracts were analyzed by GC–MS and a total of thirty alkaloids were detected (Table 2 and Table 3; Figure 1 and Figure 2). The amounts available in Table 2 are expressed as μg of galanthamine (Gal), which was related to mg of dry weight (DW). Among the observed compounds, twenty-eight were properly identified, whereas two (Unknown I and Unknown II) had not been described yet and hence were classified as miscellaneous (Table 2 and Table 3).

The alkaloid content of the Facultative Autumn-Flowering *Narcissus* (FAF) and the Strictly Autumn-Flowering ones (SAF) differed in several aspects. FAF did not have any homolycorine-type alkaloid, whereas SAF did not express tazettine, narciclasine, or haemanthamine-type alkaloids (Figure 3). Given that the biosynthetic route of the first one (*ortho-para*’ coupling) differs from the other three (*para-para*’ coupling) [7], the differences might be showing an evolutionary divergence from SAF and FAF. Furthermore, despite having common alkaloid types (lycorine, galanthamine, and miscellaneous), which comprise a total of twenty-one alkaloids, only three of them are shared by both groups (lycorine, 11,12-didehydroanhydrolycorine, and galanthamine). The fact that FAF expresses more alkaloid types than SAF could be due to their wider seasonality and environmental exposition spectra, which might require them to synthesize a vaster range of alkaloids to face them.

Only 17% of the alkaloids (6.5% of the total alkaloids amount µg Gal/mg DW) did not correspond to any Amaryllidaceae-type and were, except for *N. elegans* from Mallorca, accompanied by a much higher proportion by Amaryllidaceae alkaloids. Ismine was detected in *N. blancoi*, and hordenine in both *N. elegans* and *N. cavallinesii* (which also expressed *N*-methyltyramine). The species *N. blancoi* also showed alkaloids as narciclasine, an isocarbostyril constituent of the Amaryllidaceae family with significant antitumoral proprieties [13]. Additionally, an unidentified component was detected in *N. obsoletus*, and the other in *N. deficiens*. Indeed, it is unusual to locate non-Amaryllidaceae alkaloids in the Amaryllidoideae subfamily, and when found they are commonly accompanied by several true Amaryllidaceae alkaloids [11].

Interestingly, the bulbs of *N. obsoletus* presented a large quantity of galanthamine-type alkaloids (52% of the total amount), among which galanthamine represented 63% of the total amount. Therefore, this daffodil is an extremely interesting plant which could potentially be exploited to improve galanthamine production.

It should also be noted that narseronine corresponds to 85% of the total alkaloid content of *N. deficiens* (293.43 µg Gal/mg DW) (Figure 4). Although in less quantity, it is also present in *N. serotinus*, *N. viridiflorus*, *N. papyraceus*, and *N. obsoletus*. Narseronine is a particular homolycorine-type alkaloid that lacks a double bond between C3 and C4. It was properly characterized for the first time by Pigni et al., 2010 [14], and its biological and pharmacological properties have not been investigated yet. To do so, provided the large amount of narseronine that *N. deficiens* expresses, the plant could be used as a raw molecule source for performing those studies in the future.

### 2.2. Acetylcholinesterase Inhibition Activity

Seven out of the ten extracts of the daffodils were able to inhibit AChE (Figure 5). Regarding the IC_50_ of the extracts of SAF and FAF, no statistical difference has been found (Table 4).

Consistently with the high galanthamine content (32.62 µg Gal/mg DW), the IC_50_ value of *N. obsoletus* was the lowest of the eight daffodils studied. Indeed, its inhibition activity was considerably strong (IC_50_ = 0.92 ± 0.06 mg/mL). The plant could therefore be used either as a new source of galanthamine production or as a tool for its improvement (through techniques such as metabolic or genetic engineering, plant transgenesis, or breeding processes). However, before aiming for those goals, it should be studied more in-depth. For instance, the alkaloid content could be evaluated during different life cycle stages, conditions, and plant parts. Further, the regulatory mechanisms that affect the galanthamine content might be investigated.

Galanthamine was also found within the extract of *N. blancoi* (3.29 µg/mg DW), which was the fourth most potent inhibitor of AChE (IC_50_ of 30.25 ± 1.25mg DW/mL). This plant could also be employed for breeding processes or biotechnological improvements of galanthamine production. The sample *N. papyraceus* presented the second highest AChE inhibitory activity (IC_50_ = 8.09 ± 0.58 mg DW/mL) following *N. obsoletus* (IC_50_ = 0.92 ± 0.06 mg DW/mL). Although other studies have shown that *N. papyraceus* synthetizes galanthamine [15], the present sample did not contain it. This could be due to several reasons: for instance, the alkaloid content of the daffodil is five times higher in the aerial part than in the bulbs [16]. Hence, galanthamine could be found within the plant section that has not been analyzed in this study. It might also be explained by environmental motives or because the daffodils employed here, and the ones used by Tarakemeh et al. [15] correspond to different subspecies. Regardless of this, given that *N. papyraceus* contains 3.28 µg Gal/mg DW of assoanine and that the IC_50_ of this alkaloid has been determined to be 3.87 µM (approximately twice the IC_50_ of galanthamine = 1.9 µM) [17], the inhibitory activity of this daffodil extract might be probably due to its assoanine content.

*N. deficiens* extract was the third most active against AChE. It contained homolycorine (with no AChE inhibition activity) and galanthine (IC_50_ = 63.1 µM) [18]. Providing that galanthine amount is 3.39 µg Gal/mg DW, neither of those alkaloids could be responsible for the AChE inhibition experimentally observed. Hence, the activity should be due to at least one or a combination of the remaining components of the extract whose inhibitory activity has not been evaluated yet (narseronine, 1-*O*-acetyl-3-*O*-methylnarcissidine, 3-*O*-methylnarcissidine, 1-*O*-acetyl-3-*O*-methyl-6-oxonarcissidine, 2-*O*-methylclivonine, and an unidentified alkaloid, Unknown II).

In line with their low alkaloid content (Table 2), both *N. viridiflorus* and *N. elegans* did not show any AChE inhibitory activity, with IC_50_ of 2079.74 ± 19.03; 3003.67 ± 2.32; and 2871.11 ± 19.41 mg DW/mL, respectively. Consequently, neither narseronine, nor hordenine and 1-*O*-acetyl-3-*O*-methylnarcissidine have a strong AChE inhibitory effect.

Different Amaryllidaceae genera have also shown some anti-cholinesterase potential. For instance, *Rauhia multiflora* presented high activity against AChE, with IC_50_ values of 0.17 ± 0.02 μg/mL [19]. Additionally, new Amaryllidaceae alkaloids from *Crinum jagus* have recently been reported [20]. Although their activity was not as significant as galanthamine and sanguinine, they demonstrated a slight inhibition acetylcholinesterase [20]. Furthermore, synergistic interaction of Amaryllidaceae alkaloids in plant extracts has also been reported for the inhibition of acetylcholinesterase activity [21].

## 3. Materials and Methods

### 3.1. Plant Material

The bulbs of ten natural population of *Narcissus* grown on different regions of Andalucía and Mallorca (both Spain), and Tlemcen (Algeria) have been collected during late autumn or early winter (Table 5 and Figure A1). Plant samples were obtained and identified by the biologists Pedro Gómez-Murillo and Carles Jiménez, both specialized in this genus.

### 3.2. Alkaloid Extraction

The samples were cut into thin fragments, dried in a forced convection oven at 40 °C for 48 h, and ground up into fine powder with a rotary blade mill (stainless steel grinder, Taurus). Afterward, 300 mg of plant material were macerated with 500 µL of MeOH and 500 µL of MeOH diluted codeine (0.1 mg/mL) which served as an internal standard. The mix was then sonicated for 2 h (15 min of sonication and 15 min rest intervals), and the solvent was evaporated to dryness. Thereafter, extracts were acidified with 500 µL of H_2_SO_4_ (2%, *v*/*v*). The neutral material was removed twice with 500 µL of Et_2_O, and the rest was basified with 200 µL NH_4_OH (25%, *v*/*v*). Afterward, 500 µL of Et_2_O were added to separate the organic phase (which contained the alkaloids), three times, and the solvent was evaporated to dryness. Then, the extract was reconstituted with 200 µL of methanol and subjected to a GC–MS analysis [22].

### 3.3. GC–MS Analysis

The alkaloids were analyzed by a GC–MS carried out on the “Agilent Technologies 6890N” gas chromatographer, coupled with the “Agilent Technologies 5975 inert” mass detector operating on EI mode, and using the autoinjector “7683B Series Agilent Technologies”. The oven ramp temperature used was the following: 55–60 °C at 2.5 °C/min, 60–100 °C at 40 °C/min, 100–180 °C at 8 °C/min, and 180–300 °C at 6 °C/min. The injector temperature was 250 °C, operating on the Pulsed Spitless mode. The capillary column used was TEKNOKROMA TR-45232 Sapiens-X5MS (30 m × 250 µm ID × 0.25 µm). The interphase detector temperature was 280 °C, and the injection volume was of 1 µL.

### 3.4. Alkaloid Identification and Quantification

The alkaloids were identified by comparing their GC–MS spectra and Kovats retention indices (RI) with those from the Amaryllidaceae alkaloids library database of the Natural Products Group of the University of Barcelona (Spain). The library has been updated on a regular basis with Amaryllidaceae alkaloids isolated and unequivocally identified via spectrometric and spectroscopic methods. An extensive literature database about MS and diagnostic mass peaks has also been consulted [23,24,25,26,27]. The mass spectra were deconvoluted using the AMDIS (Automatic Mass spectral Deconvolution and Identification System) 2.64 Software.

To perform the alkaloids quantification of each extract, a galanthamine calibration curve was prepared using concentrations of 10, 20, 40, 60, 80, and 100 µg/mL. Five hundred µL of codeine (0.1 mg/mL) was added to each concentration as an internal standard. Every peak area was manually quantified, taking into consideration picked ions for each alkaloid (generally, the base peak of their MS—for galanthamine *m*/*z* = 286, and for codeine *m*/*z* = 299). In order to obtain a calibration curve, the galanthamine area/codeine area ratio was determined for each solution and plotted against the respective galanthamine concentration. The data acquired from every sample was normalized to the area of their internal standard (codeine). Consequently, the approximate amount of every alkaloid was determined using the calibration curve and equation. The resulting values were expressed as galanthamine equivalents (mg GAL), which were finally linked to the mg of DW of each plant. Even though the method does not enable absolute quantification, it allows for obtaining data for relative comparison purposes between samples.

### 3.5. AChE Inhibitory Activity

The AChE inhibitory activity was carried out according to Ellman et al. 1961. [28], with the modifications incorporated by López et al. 2002 [29]. A stock solution of 518U of AChE from *Electrophorus electricus* (Sigma, Schnelldorf, Bayern, Germany) was prepared and kept at −20 °C. Acetylcholine iodide and DTNB were also obtained from Sigma, Schnelldorf, Bayern, Germany.

To perform the analysis, 50 µL of AChE in phosphate buffer (8 mM K_2_HPO_4_, 2.3 mM NaHPO_4_, and 0.15 M NaCl, at pH 7.5) was incorporated into 50 µL of the sample dissolved in that same buffer, in flat-bottom 96-well plates. The plates were subsequently incubated for 30 min at 21–25 °C. Immediately afterwards, 100 µL of substrate solution was added (0.1 M Na_2_HPO_4_, 0.5 M DTNB, and 0.6 mM ACh iodide in Millipore water at a pH of 7.5). After 5 min, the absorbance was read on an ELISA plate reader (Multiscan EX Thermo Scientific^®^, Waltham, MA, USA). Galanthamine was used as a positive control (employing a 1:10 dilution bank from 10^−3^ to 10^−7^ M). Phosphate buffer was used as blank.

To determine the inhibitory potential of each extract, their respective IC_50_ was established. In order to do so, a semilogarithmic linear regression study was carried out from the concentrations analyzed and the inhibitory values obtained.

### 3.6. Statistical Analysis

The statistical analyses were performed using the Statgraphics Centurion 18 ^®^ software as well as the Microsoft Excel 2016 software. The results were expressed as the arithmetic median ± the standard deviation. The comparison between groups was established with a one-way ANOVA test and the statistical significance was considered as *p* < 0.05.

## 4. Conclusions

Every daffodil was able to synthesize alkaloids, most of which had known biological properties (mainly having neurological and cytotoxic effects) but some of them have not been studied yet.

There was a clear difference between the alkaloid types synthesized by SAF and FAF, possibly due to an evolutive divergence. Nevertheless, no significant difference has been found in their ability to inhibit AChE. Indeed, every daffodil extract inhibited the enzyme except for *N. viridiflorus* and *N. elegans*. An interesting case was that of *N. deficiens*, with the third most active extract, which did not express any alkaloid with known AChE inhibitory activity. Consequently, those compounds constitute potential new treatments for cholinergic disorders and should be studied more in-depth. Furthermore, one of them, narseronine, was synthesized in an enormous amount by *N. deficiens*. Hence, this daffodil could be used as a source for the isolation and study of this molecule.

Lastly, both *N. obsoletus* and *N. blancoi* synthesized galanthamine and could be used as tools for improving the production of this molecule. For instance, their unusual flowering times could be employed in breeding or genetic-engineering processes to extend the current anthesis time of the plants used for galanthamine production, thereby benefiting not only the pharmaceutical but the flowering and/or cosmetic markets.

In conclusion, even though there is little information about the autumn-flowering species of *Narcissus*, the present study suggests that they could have enormous future potential. However, it should studied more in-depth in order to exploit it.

## Figures and Tables

**Figure 1 molecules-28-01239-f001:**
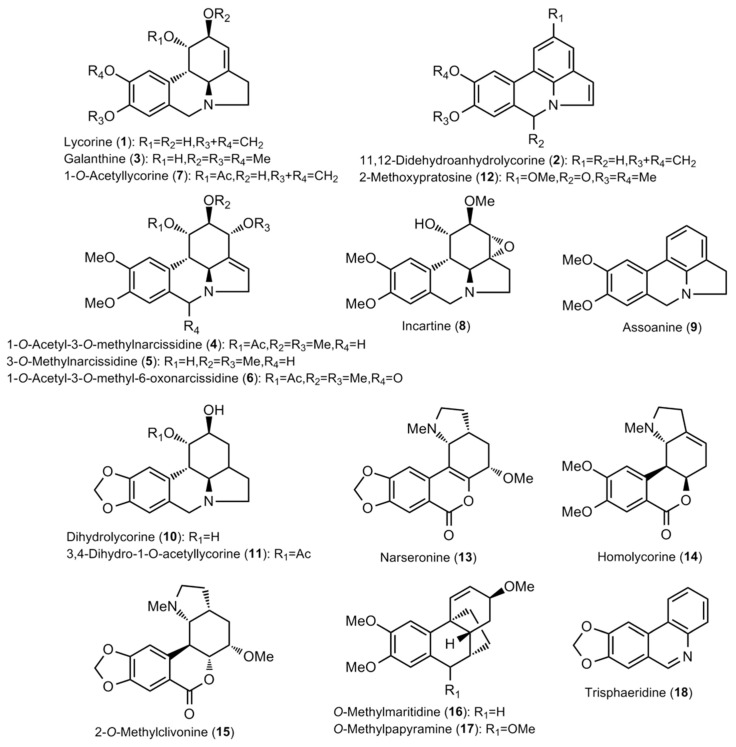
Structure of the Amaryllidaceae alkaloids found in the *Narcissus* extracts (I).

**Figure 2 molecules-28-01239-f002:**
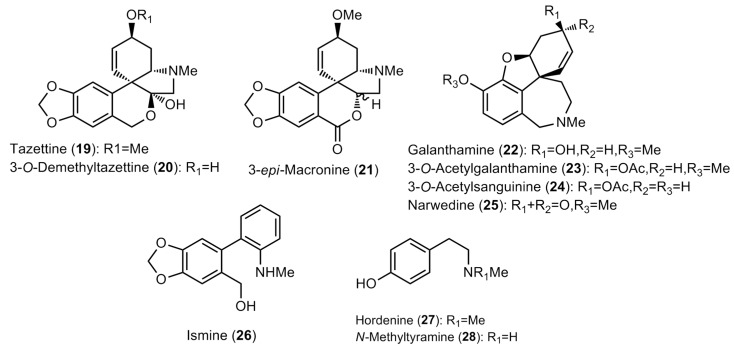
Structure of the Amaryllidaceae alkaloids found in the *Narcissus* extracts (II).

**Figure 3 molecules-28-01239-f003:**
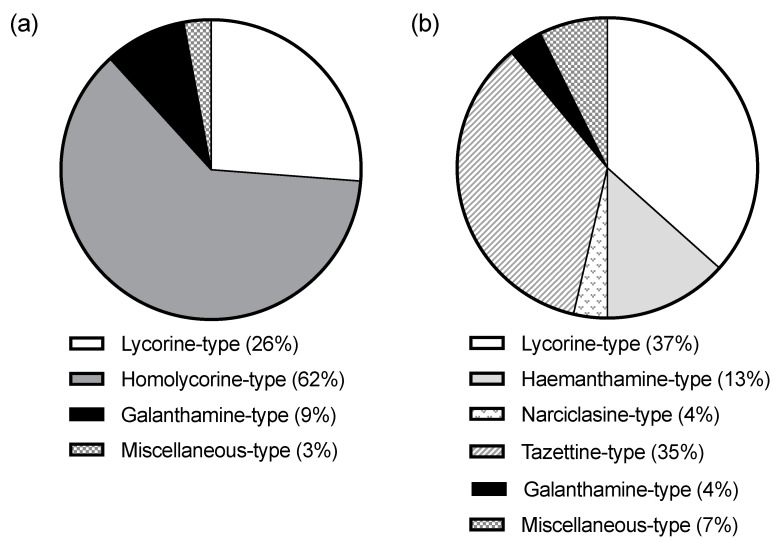
Circle graph of the alkaloid-type content of SAF (*w*/*w* %) (**a**) and FAF (*w*/*w* %) (**b**).

**Figure 4 molecules-28-01239-f004:**
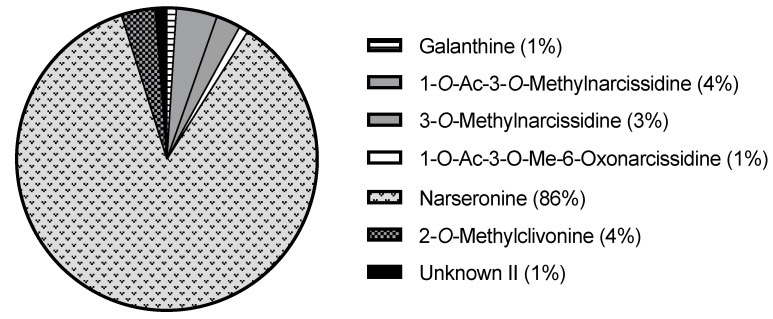
Circle graph of the alkaloid content of *N. deficiens* (*w*/*w* %).

**Figure 5 molecules-28-01239-f005:**
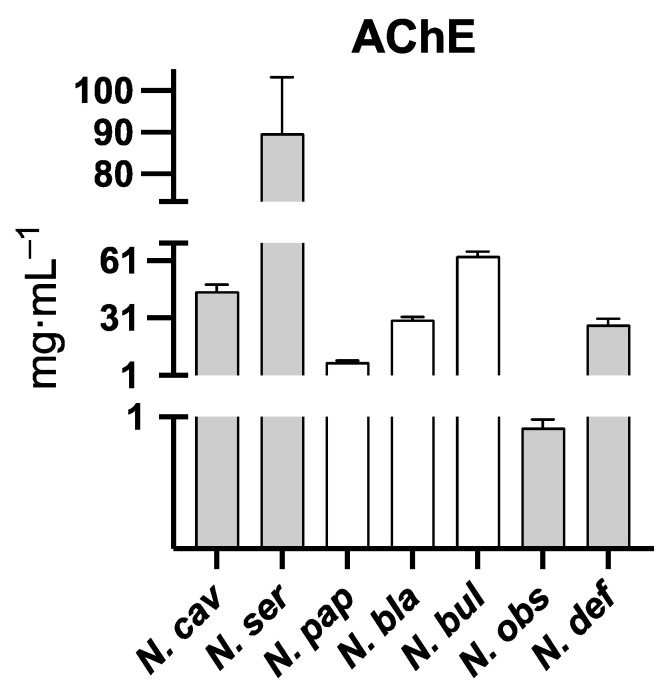
Barr plot of the AChE inhibition activity of active extracts. Respective IC_50_ values of active plant extracts are: *N. obs*: *N. obsoletus* (0.92 ± 0.06 mg DW/mL), *N. pap*: *N. papyraceus* (8.09 ± 0.58 mg DW/mL), *N. def: N. deficiens* (27.86 ± 2.93 mg DW/mL), *N. bla*: *N. blancoi* (30.25 ± 1.25 mg DW/mL), *N. cav*: *N. cavallinesii* (44.93 ± 3.47 mg DW/mL), *N. bul*: *N. bulbocodium* (63.43 ± 2.31 mg DW/mL), and *N. ser*: *N. serotinus* (89.79 ± 13.44 mg DW/mL). SAF are represented with gray bars and FAF with white ones.

**Table 1 molecules-28-01239-t001:** Registered anthesis months of the autumn-flowering daffodils that have been studied.

		Anthesis Months											
	MAT *	I	II	III	IV	V	VI	VII	VIII	IX	X	XI	XII
*N. obsoletus*	09/10												
*N. deficiens*	21/10												
*N. serotinus*	23/10												
*N. cavanillesii*	27/10												
*N. viridiflorus*	28/10												
*N. elegans ***	19/10												
*N. papyraceus*	22/01												
*N. bulbocodium*	11/02												
*N. blancoi*	17/01												

The colored labels correspond to the months where the anthesis has been registered (Pink—Spring: from 20th March to 21st June, Orange—Autumn: from 23rd September to 21st December, Blue—Winter: from 21st December to 20th March Yellow—Summer: from 21st June to 23rd September). The samples were collected from 23rd November of 2016 to 16th June of 2022 [12]. * Mean Anthesis Time (MAT). ** Two different populations of *N. elegans* have been studied, from Mallorca (Spain) and Algeria, respectively.

**Table 2 molecules-28-01239-t002:** Quantification of the alkaloids identified by GC–MS in each *Narcissus* extract. The data is expressed in µg Gal/mg DW.

	*SAF*							*FAF*		
Alkaloids	*N. viridiflorus*	*N. serotinus*	*N. cavanillesii*	*N. obsoletus*	*N. elegans (M)*	*N. elegans (A)*	*N. deficiens*	*N. blancoi*	*N. bulbocodium*	*N. papyraceus*
** * Lycorine type: * **	**-**	**108.66**	**6.58**	**8.37**	**-**	**12.72**	**30.74**	**9.96**	**-**	**23.02**
Lycorine **(1)**	-	-	3.30	4.95	-	-	-	-	-	19.51
11,12-Didehydroanhydrolycorine **(2)**	-	-	-	3.42	-	-	-	-	-	3.51
Galanthine **(3)**	-	3.48	-	-	-	-	3.39	-	-	-
1-*O*-Acetyl-3-*O*-methylnarcissidine **(4)**	-	64.29	-	-	-	12.72	15.01	-	-	-
3-*O*-Methylnarcissidine **(5)**	-	27.10	-	-	-	-	9.04	-	-	-
1-*O*-Acetyl-3-*O*-methyl-6-oxonarcissidine **(6)**	-	3.69	-	-	-	-	3.30	-	-	-
1-*O*-Acetyllycorine **(7)**	-	3.37	-	-	-	-	-	-	-	-
Incartine **(8)**	-	3.31	-	-	-	-	-	-	-	-
Assoanine **(9)**	-	-	3.28	-	-	-	-	-	-	-
Dihydrolycorine **(10)**	-	-	-	-	-	-	-	3.29	-	-
3,4-Dihydro-1-acetyllycorine **(11)**	-	-	-	-	-	-	-	6.67	-	-
2-Methoxypratosine **(12)**	-	3.42	-	-	-	-	-	-	-	-
** * Homolycorine type: * **	**3.40**	**16.29**	**3.28**	**36.39**	**-**	**-**	**305.90**	**-**	**-**	**-**
Narseronine **(13)**	3.40	10.13	3.28	3.30	-	-	293.43	-	-	-
Homolycorine **(14)**	-	-	-	33.09	-	-	-	-	-	-
2-*O*-Methylclivonine **(15)**	-	6.16	-	-	-	-	12.47	-	-	-
** * Haemanthamine type: * **	**-**	**-**	**-**	**-**	**-**	**-**	**-**	**-**	**-**	**12.07**
*O*-Methylmaritidine **(16)**	-	-	-	-	-	-	-	-	-	7.95
*O*-Methylpapyramine **(17)**	-	-	-	-	-	-	-	-	-	4.12
** * Narciclasine type: * **	**-**	**-**	**-**	**-**	**-**	**-**	**-**	**3.28**	**-**	**-**
Trisphaeridine **(18)**	-	-	-	-	-	-	-	3.28	-	-
** * Tazettine type: * **	**-**	**-**	**-**	**-**	**-**	**-**	**-**	**11.67**	**8.36**	**11.80**
Tazettine **(19)**	-	-	-	-	-	-	-	8.19	5.08	8.46
3-*O*-Demethyltazettine **(20)**	-	-	-	-	-	-	-	3.48	-	-
3-*epi*-Macronine **(21)**	-	-	-	-	-	-	-	-	3.28	3.34
** * Galanthamine type: * **	**-**	**-**	**-**	**52.03**	**-**	**-**	**-**	**3.29**	**-**	**-**
Galanthamine **(22)**	-	-	-	32.62	-	-	-	3.29	-	-
3-*O*-Acetylgalanthamine **(23)**	-	-	-	5.39	-	-	-	-	-	-
3-*O*-Acetylsanguinine **(24)**	-	-	-	3.53	-	-	-	-	-	-
Narwedine **(25)**	-	-	-	10.49	-	-	-	-	-	-
** * Miscellaneous type * ** ** /*Unknown:* **	**-**	**-**	**9.62**	**3.39**	**11.90**	**10.67**	**4.22**	**6.65**	**-**	**-**
Ismine **(26)**	-	-	-	-	-	-	-	6.65	-	-
Hordenine **(27)**	-	-	4.85	-	11.90	10.67	-	-	-	-
*N*-Methyltyramine **(28)**	-	-	4.77	-	-	-	-	-	-	-
Unknown I	-	-	-	3.39	-	-	-	-	-	-
Unknown II (homolycorine type)	-	-	-	-	-	-	4.22	-	-	-

**Table 3 molecules-28-01239-t003:** MS fragmentation by GC-EIMS of the identified alkaloids.

ALKALOIDS	RI	[M]^+^	*m*/*z*
** * Lycorine type: * **			
Lycorine **(1)**	2790.9	287 (31)	286 (19), 268 (24), 250 (15), 227 (79), 226 (100), 211 (7), 147 (15)
11,12-Didehydroanhydrolycorine **(2)**	2655.4	249 (60)	248 (100), 191 (10), 190 (24), 189 (7), 163 (7), 95 (17)
Galanthine **(3)**	2731.6	317 (26)	284 (22), 266 (8), 252 (4), 243 (79), 242 (100)
1-*O*-Acetyl-3-*O*-methylnarcissidine **(4)**	2768.5	389 (3)	388 (5), 357 (50), 326 (98), 314 (3), 298 (35), 294 (20), 284 (9), 272 (19), 266 (100), 258 (31)
3-*O*-Methylnarcissidine **(5)**	2843.8	347 (8)	348 (2), 346 (16), 315 (47), 298 (6), 284 (100), 266 (35), 258 (22), 242 (8), 230 (38), 228 (30)
1-*O*-Acetyl-3-*O*-methyl-6-oxonarcissidine **(6)**	3079.5	403 (1)	371 (10), 340 (19), 312 (4), 298 (17), 280 (100), 272 (8), 255 (10)
1-*O*-Acetyllycorine **(7)**	2750.7	329 (31)	268 (31), 250 (20), 226 (100), 211 (6), 192 (3), 167 (3), 147 (6)
Incartine **(8)**	2778.9	333 (30)	332 (77), 259 (72), 258 (100), 244 (18)
Assoanine **(9)**	2648.4	267 (52)	266 (100), 250 (31), 222 (14), 207 (14), 193 (12), 180 (15)
Dihydrolycorine **(10)**	2805.5	288 (100)	272 (7), 254 (3), 214 (6), 200 (2), 187 (6), 162 (6), 147 (15)
3,4-Dihydro-1-acetyllycorine **(11)**	2880.0	331 (41)	330 (100), 270 (28), 254 (30), 226 (10), 147 (12), 119 (7), 89 (5)
2—Methoxypratosine **(12)**	3059.0	309 (100)	294 (16), 278 (2), 266 (22), 251 (12), 236 (7), 222 (5), 208 (7), 193 (4), 164 (4), 125 (2)
** * Homolycorine type: * **			
Narseronine **(13)**	2957.5	329 (20)	299 (28), 272 (38), 256 (46), 254 (30), 242 (34), 241 (98), 240 (100), 59 (42), 57 (60), 44 (37)
Homolycorine **(14)**	2787.3	315 (<1)	206 (<1), 178 (2), 109 (100), 150 (1), 108 (22), 94 (3), 82 (3)
2-*O*-Methylclivonine **(15)**	2927.1	331 (13)	316 (6), 162 (3), 134 (2), 126 (2), 115 (2), 96 (39), 83 (100)
** * Haemanthamine type: * **			
*O*-Methylmaritidine **(16)**	2506.3	301 (42)	286 (24), 270 (22), 231 (100), 203 (26)
*O*-Methylpapyramine **(17)**	2544.3	331 (50)	300 (40), 276 (100), 245 (60), 214 (20), 201 (10)
** * Narciclasine type: * **			
Trisphaeridine **(18)**	2356.6	223 (100)	222 (38), 167 (8), 165 (9), 164 (14), 138 (20), 137 (9), 111 (13)
** * Tazettine type: * **			
Tazettine **(19)**	2682.9	331 (31)	316 (15), 298 (23), 247 (100), 230 (12), 201 (15), 181 (11), 152 (7)
3-*O*-Demethyltazettine **(20)**	2748.8	317 (24)	298 (2), 247 (100), 230 (12), 71 (42), 70 (52)
3-*epi*-Macronine **(21)**	2832.3	329 (27)	314 (23), 245 (100), 225 (14), 201 (83), 139 (16), 70 (18)
** * Galanthamine type: * **			
Galanthamine **(22)**	2442.8	287 (83)	286 (100), 270 (13), 244 (24), 230 (12), 216 (33), 174 (27), 115 (12)
3-*O*-Acetylgalanthamine **(23)**	2573.5	329 (25)	270 (100), 216 (20), 165 (15),115 (10)
3-*O*-Acetylsanguinine **(24)**	2623.1	315 (37)	256 (100), 255 (42), 254 (37), 212 (26), 165 (33), 152 (23), 115 (30), 96 (67)
Narwedine **(25)**	2517.3	285 (91)	284 (100), 242 (18), 216 (20), 199 (18), 174 (31), 128 (16), 115 (16)
** * Miscellaneous type * /*Unknown:* **			
Ismine **(26)**	2273.0	257 (28)	238 (100), 211 (6), 196 (8), 168 (6), 154 (3), 106 (4), 77 (3)
Hordenine **(27)**	1559.7	165 (<1)	121 (4), 107 (16), 91 (5), 77 (14), 58 (100)
*N*-Methyltyramine **(28)**	1550.6	151 (40)	120 (15), 108 (42), 107 (100), 91 (26), 77 (90), 65 (25)
Unknown I	2872.5	-	331 (10), 302 (100), 270 (10), 241 (10), 229 (10), 211 (10), 128 (10), 115 (10)
Unknown II (homolycorine type)	2724.3	-	109 (100)

RI = Kovats Retention Index.

**Table 4 molecules-28-01239-t004:** Analysis of variance test of the IC_50_ values of SAF and FAF.

Source	Sum of Squares	DF	Mean Square	*F*-Value	*p*-Value
Between groups	704.018	1	704.018	0.75	0.393
Intra groups	26185.3	28	935.189		
Total	26889.3	29			

Homoscedasticity between groups has been previously confirmed through Levane’s test (“*p* = 0.3782”).

**Table 5 molecules-28-01239-t005:** Plant material site and date of collection.

Plant species	Origin	Date
*Narcissus cavanillesii*	Morón de la Frontera (Sevilla)	18 October 2021
*Narcissus obsoletus*	Villanueva de Cauche (Málaga)	18 October 2021
*Narcissus deficiens*	Morón de la Frontera (Sevilla)	18 October 2021
*Narcissus serotinus*	Morón de la Frontera (Sevilla)	18 October 2021
*Narcissus bulbocodium*	Los Barrios (Cádiz)	13 January 2022
*Narcissus blancoi*	Vilches (Jaén)	17 March 2022
*Narcissus papyraceus*	Los Barrios (Cádiz)	13 January 2022
*Narcissus viridiflorus*	Facinas (Cádiz)	8 November 2021
*Narcissus elegans*	Mallorca (Balearic Islands)	2 September 2022
*Narcissus elegans*	Tlemcen (Algeria)	3 October 2022

## Data Availability

Not applicable.

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
