# Peer review of "Alkaloid Profile in Wild Autumn-Flowering Daffodils and Their Acetylcholinesterase Inhibitory Activity"

_molecules, 2023, doi:10.3390/molecules28031239_

Round 1

Reviewer 1 Report

The manuscript “Alkaloid profile in wild autumn-flowering daffodils and their acetylcholinesterase inhibitory activity" is devoted to the study of acetylcholinesterase inhibitory activity of alkaloids from genus Narcissus species. Alkaloids were identified with GS-MS and quantified using galanthamine equivalent. Nine Narcissus species was studied. The data obtained can be useful for expanding the understanding of the biological activity of secondary metabolites of plants of the genus under consideration.

I think, this manuscript can be published in the Molecules after minor revision after taking into account general recommendation and some of specific comments given below:

1.      Figures: error bars should be added.

2.      Coordinates of plant material collection and dates of collection should be presented.

3.      What standards for the preparation and storage of plant raw materials were used in the work? What particle size was obtained after grinding?

4.      On the basis of what works or own experiments was the procedure of extraction of alkaloids chosen?

5.      How many repetitions of extraction and analysis were used?

Reviewer 2 Report

see attached file

Round 2

Reviewer 2 Report

The answer on two points (extraction methods used and quantification of alkaloids) are not full satisfy the cirticisms. THese two pint shouf be further revised 
